# Valuing Provision Scenarios of Coastal Ecosystem Services: The Case of Boat Ramp Closures Due to Harmful Algae Blooms in Florida †

**Sergio Alvarez [1],\*, Frank Lupi [2] , Daniel Solís [3] and Michael Thomas [3]**

1   Rosen College of Hospitality Management, University of Central Florida, Orlando, FL 32819, USA
2   Department of Agricultural, Food, and Resource Economics, Michigan State University, East Lansing, MI 48824, USA; lupi@msu.edu
3   College of Agriculture and Food Sciences, Florida A&M University, Tallahassee, FL 32307, USA; daniel.solis@famu.edu (D.S.); michael.thomas@famu.edu (M.T.)
\*   Correspondence: sergio.alvarez@ucf.edu; Tel.: +1-407-903-8001
†   Authorship ordered alphabetically, no senior authorship implied.

**Abstract:** Ecosystem service flows may change or disappear temporarily or permanently as a result of environmental changes or ecological disturbances. In coastal areas, ecological disturbances caused by toxin-producing harmful algae blooms can impact flows of ecosystem services, particularly provisioning (e.g., seafood harvesting) and cultural services (e.g., recreation). This study uses a random utility model of recreational boating choices to simulate changes in the value of cultural ecosystem services provided by recreation in coastal ecosystems resulting from prolonged ecological disturbances caused by harmful algae blooms. The empirical application relies on observed trips to 35 alternative boat access ramps in Lee County, an important marine access destination in southwest Florida. Results indicate that reduced boating access from harmful algae blooms may have resulted in losses of $3 million for the 2018 blooms, which lasted from the end of June to the end of September.

**Keywords:** harmful algae blooms; cyanobacteria; recreational boating; ecosystem services; random utility model; economic analysis

## 1. Introduction

Coastal ecosystems provide a diversity of services that contribute to social well-being. While human use and enjoyment of some of these services are captured (and measurable) by market transactions, most uses of these vital ecosystem services are not. Among these non-market ecosystem services, perhaps the most readily measurable is recreational use of waterways, particularly services related to recreational boating. Although recreational boating does not account for the total value of coastal ecosystems and the services they provide, recreational boating in Florida (FL) is an important cultural service[A] and a key component of the value of coastal ecosystem services. In 2017 there were close to 12 million registered recreational boats in the United States (US), and nearly 1 million of these were in FL [1]. These boaters enjoy the cultural services provided by clean waterways and healthy coastal ecosystems. Understanding the monetary value of these services can help coastal managers and policy-makers in their decision-making processes.

For boaters, the boat ramp infrastructure provides access to cultural ecosystem services produced in coastal waters. These ecosystem services produce economic benefits to the boaters in the form of increased well-being and satisfaction from boating, and these benefits are above and beyond the direct costs of boating (e.g., transportation costs of pulling boats to the ramp, fuel for the boats, fees, etc.). Economists refer to such benefits as consumer surplus, and this surplus comprises one side in the

benefit-cost equation in situations where the benefits of boating-related coastal ecosystems services may be weighed against the costs of protecting, conserving, or restoring the ecosystems that produce them.

In this study, we develop a model that estimates changes in the consumer surplus from boating to inform a policy-relevant scenario related to ecological disturbances from harmful algae blooms (HABs). Specifically, the economic models developed here are models of the demand for access to boating sites and are suitable for valuing access as well as the characteristics of boating sites. We use random utility models (RUMs) informed by data on individual trips to explain boaters' site choices and to relate these choices to the costs of access and characteristics of alternative boating sites [2]. Boaters' choices reveal their relative preferences for site characteristics and travel costs, i.e., the boaters' willingness to trade costs (or money) for site characteristics or specific destinations. Through this linkage, RUMs can value changes in access to specific sites or individual characteristics, and they can be used to measure the welfare implications of changes in the provision and quality of ecosystem services. Similar models have been successfully employed to measure these changes for a wide variety of coastal ecosystem services [3–7]. As mentioned by [8], the model contributes to the relatively thin existing literature valuing recreational boating.

In this article, we use data and estimates from a baseline study and data collection commissioned by the FL Fish and Wildlife Conservation Commission [9]. A RUM model is used to simulate losses in consumer surplus as a result of recent and ongoing harmful algae blooms (HABs) of cyanobacteria from the genus *Microcystis* and *Anabaena*, which besides resulting in unsightly green tinted water and foul smells also produce toxins that lead to fish kills and adverse impacts to human health [10,11]. To accomplish this, we develop scenarios of ramp closures as a result of these HABs, and we examine the impact of duration of the algae blooms on the loss of consumer surplus from boating. This study contributes to thin literature on economic valuation of HABs on ecosystem services [12]. In addition to presenting the boating RUM model results, the paper illustrates an approach for using economic valuation studies to inform policy-relevant scenarios of environmental degradation that arise after the initial study is commissioned.

The rest of this article is organized as follows. Section 2 introduces the problem of HABs in Florida, specifically in the study region, and discusses how this problem impacts the ecosystem services enjoyed by recreational boaters. Section 3 outlines the methodology, and Section 4 presents the data used to develop the RUM. Section 5 provides an overview of the results of the empirical application as well as the evaluation of closure scenarios to provide a tangible application for our framework. The paper ends with a discussion and conclusion in Sections 6 and 7.

## 2. Harmful Algae Blooms and Their Impact on Recreational Boating

Coastal and waterfront communities depend on clean water as the foundation of a healthy and growing economy. Visitors from around the world are drawn to clean and pristine beaches, lakes, and other waterways, and local communities benefit from the economic activity that occurs when visitors spend their money in hotels, restaurants, shops, and other services. Similarly, residents of waterfront communities regularly use beaches and other waterways for recreation and community-building. In short, clean water is the thread that ties waterfront communities together, drives their economies, and provides a quality of life for residents and a positive experience for visitors.

However, issues related to eutrophication and HABs are having negative consequences on social well-being in many parts of the world [13–18]. In numerous FL waterfront communities, widespread HABs are severely impacting water quality and are having a deleterious impact on local economies and human health. HABs occur when colonies of photosynthetic microorganisms that live in fresh or saltwater grow out of control and produce toxins that can have harmful effects on people or wildlife. In recent years, the state of Florida has experienced several HABs, most notably outbreaks of red tide (*Karenia brevis*) and cyanobacteria (*Microcystis spp.*, and *Anabaena spp.*).

HABs have become more frequent in recent years as a result of warmer temperatures and nutrient pollution, and threaten to become a chronic issue for many communities in FL. Not surprisingly, the

spread of these blooms has also affected the tourism industry in the state, which has a substantial impact on FL's economy. Namely, the 118.5 million people that visit FL each year provide an infusion of $111.7 billion to the state's economy [19].

The 2018 HABs provide a worrisome picture of what this problem may look like in the future. During the month of June, a significant cyanobacteria bloom emerged in Lake Okeechobee and was transported to both the southeast and southwest coasts of Florida via the St. Lucie and Caloosahatchee rivers respectively [20]. However, in southwest Florida, the situation was compounded by an unprecedented bloom of red tide that moved into the region's beaches and waterways from the Gulf of Mexico. Consequently, some communities in southwest FL experienced two different types of HABs: Cyanobacteria originating inland in Lake Okeechobee, and red tide originating offshore in the Gulf of Mexico.

These unprecedented HABs on both coasts have significantly impacted local economies, public health, and the environment. HABs are likely to have long-lasting impacts on residents' well-being [10,11] and may also damage FL's brand as a world-class tourism destination. Nationwide, eutrophication, and HABs occurring in freshwater—mostly outbreaks of cyanobacteria—cost an estimated $2.2 billion per year [21]. However, there is a notable absence of research quantifying the impact of these blooms on local economies and visitor's choices, and most of the related research that has been done in FL has focused on the impacts of red tide [22].

The cyanobacteria prevalent in FL are naturally occurring in freshwater lakes, and they are generally present in very low concentrations throughout the year. However, high nutrient loads of phosphorus and nitrogen provide the fuel that allows these cyanobacteria to reproduce into uncontrolled noxious blooms [23]. In FL, a large portion of the state's urban and rural runoff flows into the Kissimmee River Basin, where water is conveyed from densely populated central FL southward toward Lake Okeechobee. Historically, water in Lake Okeechobee would continue flowing south through the Everglades, but this historic flow has been modified through a system of canals and the construction of an earthen dike to prevent flooding in response to a 1928 hurricane that resulted in more than 2500 deaths [24]. Today, rather than flow south toward FL Bay through the Everglades, water from Lake Okeechobee flows east through the St. Lucie River into the St. Lucie Estuary, and west through the Caloosahatchee River into the Gulf of Mexico (Figure 1). Lake discharges are managed by the US Army Corps of Engineers through a system of locks and gates [25].

Cyanobacterial HABs in Lee County are fairly predictable events that originate in the nutrient-rich waters of Lake Okeechobee and are transported to the Caloosahatchee River via Lake discharges through managed locks and gates. As the cyanobacterial blooms are transported into saltwater the bacteria's cell membranes are compromised, the cells die, and any cyanotoxins in the cell are released into the water [25,26]. The cyanobacteria blooms, therefore, dissipate as they reach areas with high water salinity in the Gulf of Mexico. In recent years, cyanobacterial HABs have been observed throughout the Caloosahatchee River, but the HABs have tended to dissipate as they reach the higher salinity areas where the mouth of the river reaches the Gulf of Mexico (Figure 2).

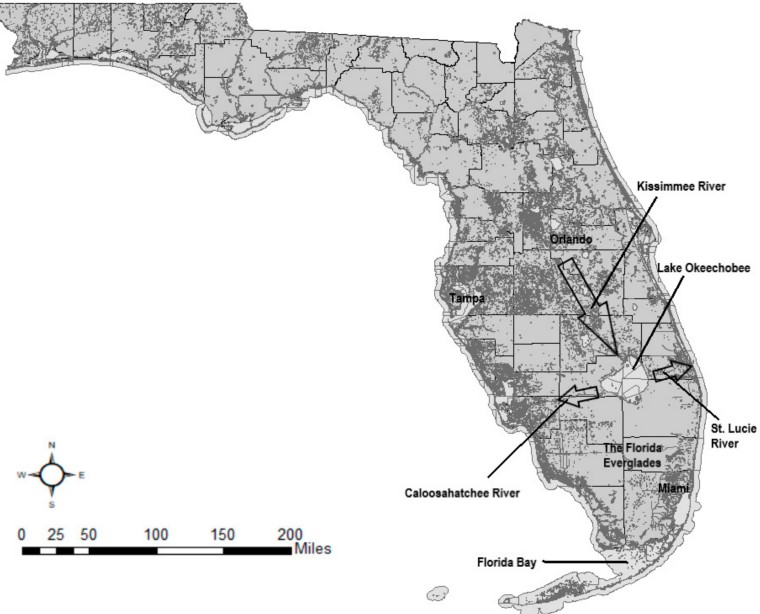

**Figure 1.** Map of Florida showing current water flow into and out of Lake Okeechobee. The locations of the densely populated cities of Orlando, Tampa, and Miami are noted, as well as the locations of the Kissimmee River, Lake Okeechobee, the Caloosahatchee River, the St. Lucie River, the Florida Everglades, and Florida Bay. Arrows show the current direction of water flow impacting the study region, which begins in the Orlando area in Central Florida and flows south into Lake Okeechobee. Prior to construction of Florida's system of water control structures and canals, water would flow south from Lake Okeechobee into the Florida Everglades, and continue flowing in a southward direction into Florida Bay. After construction of the water control system, water instead flows west to the Caloosahatchee River, and east to the St. Lucie River.

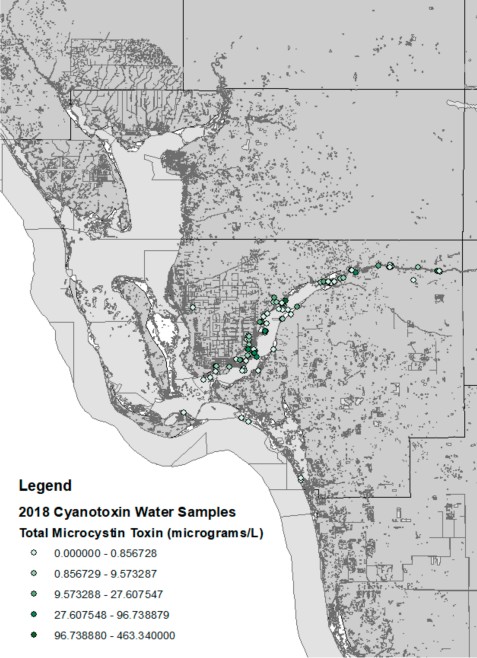

**Figure 2.** Water sampling locations and concentrations of cyanotoxins in Lee County throughout the 2018 cyanobacterial HABs. High concentrations of microcystin were observed in several locations along the Caloosahatchee River, but only very low or no concentrations were observed beyond the mouth of the river where salinity is higher [27].

## 3. A Random Utility Model to Value Site Closures Due to HABs

When estimating a model of demand for public goods such as boat ramps and beaches, the RUM approach is particularly well suited when there are many identifiable substitutes from which to choose [28,29]. This method has been widely used in the region. In the mid-1990s, the FL Department of Environmental Protection successfully used a RUM to estimate the recreational value that was lost to beach visitors following the 1993 Tampa Bay oil spill [30]. References [3–6,31–33] have applied RUMs to estimate changes in welfare resulting from perturbations in recreational fishing and boating. RUMs have also been used to evaluate the welfare lost to boaters from policies designed to protect the West Indian manatee in Lee County (restricted boating speeds and waterway access) and were later extended to Brevard County in 2003 [7,34].

In our application, it is assumed that a boater will choose a combination of a launch ramp and on-the-water destination among many possible alternatives on each choice occasion. The factors that affect choice include the cost of traveling to the ramp, the cost of boating to the desired on-the-water destination, the perceived quality of the location as a recreation site, and other characteristics of the ramp and site. We can model the individual's conditional indirect utility from visiting site $j$ as a linear function of trip costs and site characteristics given by $tc_j$ and $q_j$:

$$v_j = \beta_{tc} tc_j + \beta_q q_j + \varepsilon_j, \tag{1}$$

where $tc_j$ is the cost of traveling to the site $j$, $q_j$ is a vector of the site $j$ characteristics, $\varepsilon_j$ is a random error term accounting for factors that remain unobservable, and the $\beta$s are parameters to be estimated. The absolute value of the travel cost parameter $\beta_{tc}$ is hypothesized to be negative and serves as a measure of the marginal utility of income. The elements of vector $\beta_q$ are the marginal utilities of site characteristics and are expected to be positive if the characteristics are desirable and negative if undesirable. Following RUM theory, an individual is assumed to select the site with the highest utility. Thus, the probability of an individual choosing site $i$ is given by:

$$\Pr\left(\beta_{tc} tc_i + \beta_q q_i + \varepsilon_i > \beta_{tc} tc_j + \beta_q q_j + \varepsilon_j\right) \text{ for all } i \neq j. \tag{2}$$

Assuming the random errors are independently and identically distributed type I extreme value, Equation (2) can be estimated by a conditional logit model. In our case, we expect that the errors associated with on-the-water destinations are more correlated with one another than they are with error terms associated with the boat ramps, so we adopt a nested logit model in which the water destination sites are nested below ramp sites. Although the decision of ramp and on-the-water site is assumed to be made simultaneously, this two-level nesting structure can be modeled as an individual choosing a ramp and then choosing the water site conditional upon the selected ramp.

Let $k$ represent Lee County ramps and $j$ represent on-the-water sites, which are nested by each ramp $j$. A combination of on-the-water destination reached from a particular ramp is represented by a combination of $(j, k)$. The equations can be rewritten to reflect these nests as:

$$v_{jk} = \beta_{tc} tc_{jk} + \beta_q q_{jk} + \varepsilon_{jk} \tag{3}$$

$$\Pr\left(\beta_{tc} tc_{il} + \beta_q q_{il} + \varepsilon_{il} > \beta_{tc} tc_{jk} + \beta_q q_{jk} + \varepsilon_{jk}\right) \text{ for all } i \neq j \text{ and } l \neq k \tag{4}$$

Let $\Pr(j,k)$ be the probability of choosing site $(j,k)$ from among all feasible combinations, that is the probability that indirect utility from site $(j,k)$ exceeds the indirect utility from any other site. Assuming error terms $\varepsilon_{jk}$ are distributed as generalized extreme value, then following [35] the probability of choosing site $(j,k)$ is:

$$\Pr(j,k) = \frac{\exp\left(v_{jk}/\theta\right)\left[\sum_{j=1}^{J_k} \exp\left(v_{jk}/\theta\right)\right]^{\theta-1}}{\sum_{k=1}^{K}\left[\sum_{j=1}^{J_k} \exp\left(v_{jk}/\theta\right)\right]^{\theta}} \tag{5}$$

where $\theta$ is the nested logit distributional parameter to be estimated. To clarify our estimation approach, consider $\Pr(j, k)$ as the product of the conditional probability of choosing site $j$, given ramp $k$, $\Pr(j|k)$, times the marginal probability of choosing ramp $k$, $\Pr(k)$. That is,

$$\Pr(j,k) = \Pr(j|k)\Pr(k) = \frac{\exp(v_{jk}/\theta)}{\sum_{j=1}^{J_k} \exp(v_{jk}/\theta)} \times \frac{\left[\sum_{k=1}^{J_k} \exp(v_{jk}/\theta)\right]^\theta}{\sum_{k=1}^{K}\left[\sum_{j=1}^{J_k} \exp(v_{jk}/\theta)\right]^\theta}, \tag{6}$$

where $\Pr(k)$ and $\Pr(j|k)$ are given by:

$$\Pr(j|k) = \frac{\exp(v_{jk}/\theta)}{\sum_{j=1}^{J_k} \exp(v_{jk}/\theta)} \text{ and} \tag{7}$$

$$\Pr(k) = \frac{\left[\sum_{k=1}^{J_k} \exp(v_{jk}/\theta)\right]^\theta}{\sum_{k=1}^{K}\left[\sum_{j=1}^{J_k} \exp(v_{jk}/\theta)\right]^\theta} \tag{8}$$

another expression for $\Pr(k)$ is:

$$\Pr(k) = \frac{\exp(\theta IV_k)}{\sum_{k=1}^{K} \exp(\theta IV_k)}, \tag{9}$$

where $IV_k = \ln(\sum_{j=1}^{J_k} \exp(v_{jk}/\theta))$ is known as the inclusive value for ramp $k$ and $\theta$ is the inclusive value parameter. Note too that if the utility function contains characteristics that do not vary across water sites but do vary across ramps, which is a realistic assumption, we can re-write Equation (9) as:

$$P_{ijk} = \frac{\exp(\beta Z_k + \theta IV_{j|k})}{\sum_{k=1}^{K} \exp(\beta Z_k + \theta IV_{j|k})}, \tag{10}$$

where the $Z_k$ represent characteristics of the ramps.

In this notation, both choice probabilities take the conditional logit form. A consistent estimation strategy for the nested logit is to estimate two conditional logits linked by the lower level inclusive value index. We present the sequentially estimated model below with the first part corresponding to on-the-water site choices conditional on a ramp chosen, and the second part corresponding to the ramp choices as a function of the inclusive value of the on-the-water sites available from each ramp.

The resulting estimated model can be used for policy analysis, as the measure of welfare change estimated (benefits or damages) follows the earlier work of [36]. The welfare change resulting from the removal of sites from the choice set due to HABs can then be calculated as the consumer surplus, which, if we abstract from the nesting for the sake of exposition, is:

$$CS = -\frac{1}{\beta_{tc}}\left[\ln \sum_{k=1}^{K} e^{V_{noHAB}} - \ln \sum_{k=1}^{K} e^{V_{HAB}}\right] \times T \tag{11}$$

where $V_{noHAB}$ are the indirect utilities derived by boaters from the full choice set with $K$ ramps when there are no cyanobacterial HABs, $V_{HAB}$ are the indirect utilities derived by boaters when cyanobacterial HABs are present (which can indicate diminished utility or ramp closure), $\beta_{tc}$ is the parameter for travel cost that represents the marginal utility of money, and $T$ is the estimated number of choice occasions impacted by HABs.

### 4. Data

The first step in estimating the choice model is to define those ramps that are available to the boating public (see Supplementary Materials). In September of 2006, a field team led by the FL Fish and Wildlife Conservation Commission conducted a pilot study in Lee County that surveyed all ramp in the area. Of the 97 Lee County inventoried ramps, 55 ramps were not available for public use for a variety of reasons including temporary closures, private or gated facilities, and government ramps open only for official use. Included in the remaining 42 ramps are the obvious stand-alone public ramps and public access marinas with launch lanes. Ramps that are closed to public access were excluded from the analysis.

Next, the location of ramps in relation to each other was considered. When choosing an access point, boaters likely consider ramps in close proximity to one another as members of a larger group or aggregate. For example, if the parking lot of one ramp is full the boater could easily move along to the nearby neighboring ramp with no significant increase in travel time or cost, and still be able to reach the desired on-the-water site. Therefore, nearby ramps were aggregated as a single ramp to capture this choice behavior. Specifically, ramps within 1.5 road miles of each other were grouped and considered single aggregated ramps. For Lee County, twelve ramps were aggregated into five groups leaving a total of 34 individual ramp choices (Figure 3).

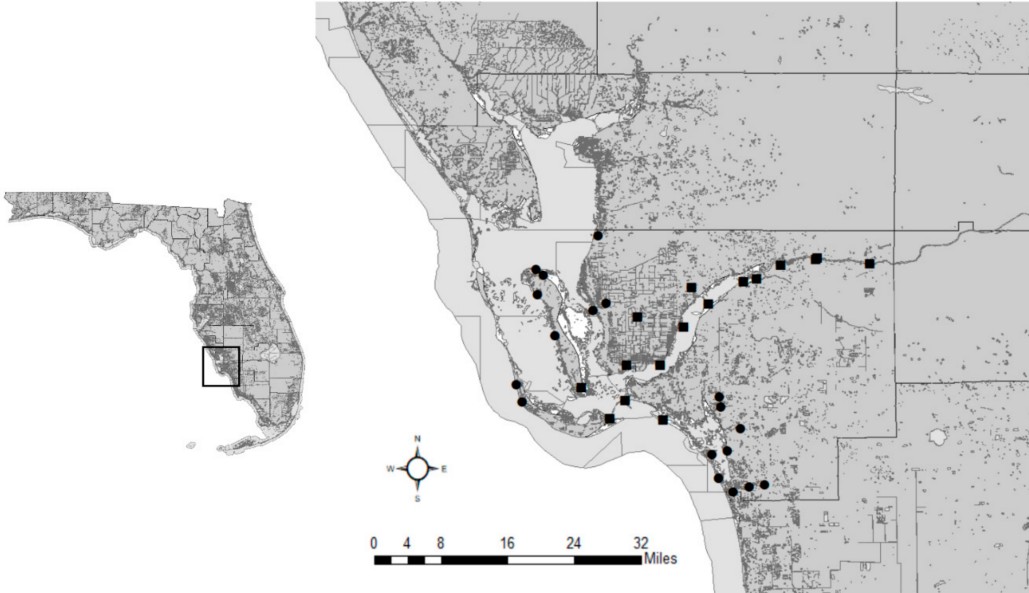

**Figure 3.** Map of Florida and location of boat ramps in Lee County. Ramps represented by squares experience cyanobacteria harmful algae blooms originating from Lake Okeechobee discharges. Boat ramps represented by circles are assumed not to be impacted by cyanobacteria algae blooms.

With the ramps selected, the next step in preparing the data involved identifying on-the-water destination sites. Florida Fish and Wildlife Conservation Commission (FWC) constructed a statewide GIS grid overlay comprised of 73,485 one-mile-square cells. Each grid cell contained at least 30 variables representing cell attributes including the presence or absence of salt and/or fresh water, natural and/or artificial reefs, seagrass, navigational aids, manatee protection status and marine protection/conservation status. Information also included bathymetry data and lake acreage, among other variables. For Lee County, the one-mile-square grid cells were aggregated into 12 square mile polygons, and cell attributes were statistically averaged for each polygon. In the boating survey, boaters were asked to identify their on-the-water destination using a geo-referenced mapping system. Their choice was then linked to the correct polygon with its aggregated site attributes. In this study, we focus on individuals taking day trips.

Statewide there were 26,771 trip-level survey responses during the 12-month sampling period. Of this number, 6690 (25%) reportedly used a boat ramp during their trip. Of those using a boat ramp, 195 (2.9%) used Lee County ramps. Some of these trips used private access (not valid for a public access model), and others failed to identify a valid boat ramp so were removed from the analysis. After adjusting for long distance trips, a total of 153 valid trips were available for the RUM analysis.

Travel costs were computed using the miles traveled from the point of origin to the boat ramp, in addition to the launch fees, which vary by ramp. The miles traveled was derived from the PC-miler software by adding the road miles from the origin of the trip to the location the boat is kept (which are the same in many cases) to the road miles from there to the latitude and longitude geolocation associated with each of the ramp groups. Travel costs were then the sum of the launch fee, bridge tolls, the driving cost assuming towing[B] and the time costs derived as the driving time (m/45 mph) multiplied by the time value (annual income/2080 h per year)[C].

Spatial analysis was conducted with the ArcGIS software, and all numerical analysis was conducted with the LIMDEP software.

## 5. Results

The estimation results for the model of water site choices, conditional upon a ramp, are presented in Table 1. The table gives the estimated parameters, their standard errors (S.E.), and the levels at which the parameters can be considered to be statistically significant (*p*-values). The overall model is statistically significant based on a chi-squared test of the joint parameter values. The coefficient of travel cost is significant and of the expected sign.

**Table 1.** Random utility model estimates for choice of water sites.

| Variable (Water Site Characteristic) | Coefficient | S.E. | *p*-Value |
| --- | --- | --- | --- |
| Travel cost | −0.4609 | 0.0452 | <0.0001 |
| Navigation aids in grid | −0.9250 | 0.4908 | 0.0595 |
| Artificial reef in grid | −5.1340 | 2.3967 | 0.0322 |
| Marine protected or conservation zone in grid | 2.1276 | 0.3721 | <0.0001 |
| Manatee zone in grid | −1.2558 | 0.4550 | 0.0058 |
| Mean depth | 0.3174 | 0.0672 | <0.0001 |
| Nearest ramp distance | −0.4411 | 0.0904 | <0.0001 |
| N = 153 | | | |
| LLF = −516.65 | | | |
| McFadden R$^2$ = 0.209 | | | |

The results indicate that the final water destinations chosen by survey respondents are less likely to be in grids with navigation aids (significant at 10% but not at 5%). Similarly, grids with artificial reefs were less likely to be selected as on-the-water destinations. Water sites with marine protected zones or with conservation zones within the grid were significantly more likely to be chosen. Alternatively, water grids with a manatee zone were significantly less likely to be selected as on-the-water destinations. The mean depth of a grid was positively associated with the choice of on-the-water destination, reflecting a preference for deeper water. Finally, the distance from the water site to the nearest ramp (defined as any ramp, not just the ramp they launched from) was negatively associated with the choice of on-the-water destinations. In sum, preferred water destinations had low travel costs, were close to a ramp and near a conservation zone, yet were in deeper water away from navigation aids, artificial reefs, and manatee zones.

The estimation results for the model of ramp site choices are presented in Table 2. The table gives the estimated parameters, their S.E., and the levels at which the parameters can be considered to be statistically significant (*p*-values). The overall model is significantly based on a chi-squared test of the joint parameter values. The travel cost for getting to the ramp is significant and of the expected

(negative) sign, implying that all else equal boaters prefer ramps that are nearer to their points of origin over ramps that are farther away.

**Table 2.** Random utility model estimates for choice of ramp groups.

| Variable (Water Site Characteristic) | Coefficient | S.E. | *p*-Value |
|---|---|---|---|
| Travel cost | −0.0299 | 0.003 | <0.0001 |
| Inclusive value of water sites | 0.4586 | 0.126 | 0.0003 |
| Number of sites within group | 0.8701 | 0.138 | <0.0001 |
| Average parking size (1000's) | 0.0328 | 0.008 | 0.0001 |
| Parking condition index | 0.8340 | 0.328 | 0.0111 |
| Ramp development index | 4.4716 | 0.618 | <0.0001 |
| Marina | −1.4790 | 0.237 | <0.0001 |
| N = 153 | | | |
| LLF = −391.25 | | | |
| McFadden $R^2$ = 0.281 | | | |

The inclusive value parameter for on-the-water sites is significant, and the parameter lies between 0 and 1, which is consistent with the theory for nested logits [2]. The parameter is also significantly different than one, which indicates the superiority of the nesting structure relative to a simple un-nested conditional logit model. The number of ramps within a group was positive and significantly different than zero. The theory of aggregation of sites with random utility models suggests that the number of elements in a group should have a parameter of one [37], and our result is consistent with the aggregation theory since the parameter on the number of ramps in a group is not significantly different from one.

The average parking size is significant and positive, as is the index of parking condition. Ramps with higher levels of development (measured by average facility counts) were significantly preferred to those with lower levels of facilities. However, marinas were less preferred by those trailering their boats to a ramp.

Table 3 presents information for the specific ramp groups. Columns five and six show the survey data on ramp choices (giving both the ramp shares and the frequencies). The seventh column presents the predicted probability of selecting a particular ramp. We can see that the model fit roughly corresponds to the distribution of the sample shares. In particular, the model predicts the highest visitation probability for the ramp with the most visits, and similarly predicts relatively high visitation for sample ramps with high visitation. Similarly, most of the ramps that received low or no visits are predicted to have low probabilities of use. Longitude and latitude coordinates for each ramp group can be found in Appendix A, Table A1.

The second column shows the access value per choice occasion for each of the ramps using the consumer surplus as a welfare measure (Equation (11)), where choice occasion means "taking a trip to a Lee County ramp." This amount represents the lost economic value to boaters if they were to lose access to the site for one choice occasion, yet retain access to the other boat ramps in Lee County. The value is in the range of others reported in the literature but is higher than values reported for access to Hawaii ramps [38]. It is important to note that the values reported in Table 3 are values that accrue to all ramp boating trips made to Lee County (i.e., the scope of choices in the model), and are not the values for a specific visitor that has visited a ramp for which access is lost, which are commonly reported in the literature. In the RUM, we can approximate such site-specific values for lost trips to a specific site by dividing the Lee County values per choice occasion for specific ramps by the probability of making a Lee County trip to that ramp [28,35]. If we make these adjustments for the trips to a particular ramp, we obtain values in the range of $30–40 per trip to a specific ramp, which are within the range of user day values found in the recreation literature.

**Table 3.** Estimated site values and observed and predicted trips to the ramp groups. Sites with closures due to HABs are indicated. Consumer surplus estimates are reported in 2009 dollars.

| Ramp Name | CS Per Choice Occasion | Closure Due to HAB | Lower Bound CS Loss | Survey Data on Ramps | | Predicted Probability a Lee County Trip is to a Particular Ramp |
| --- | --- | --- | --- | --- | --- | --- |
| | | | | Visitation Shares | Frequency | |
| BMX Strausser | $1.09 | Yes | $1.09 | 0.00% | 0 | 0.032 |
| Alva Boat Ramp | $0.20 | Yes | $0.20 | 0.00% | 0 | 0.006 |
| Burnt Store Boat Ramp | $1.99 | No | $0.00 | 5.90% | 9 | 0.059 |
| Cape Coral Yacht Basin | $1.64 | Yes | $1.64 | 5.90% | 9 | 0.048 |
| Lovers Key/Carl E. J | $2.71 | No | $0.00 | 9.20% | 14 | 0.07 |
| City of Fort Myers Yacht Basin | $1.11 | Yes | $1.11 | 6.50% | 10 | 0.033 |
| Ft Myers Shores Davis Ramp | $0.34 | Yes | $0.34 | 0.70% | 1 | 0.01 |
| Franklin Locks North | $0.28 | Yes | $0.28 | 0.00% | 0 | 0.008 |
| Franklin Locks South | $0.40 | Yes | $0.40 | 0.70% | 1 | 0.012 |
| Bokeelia Boat Ramp | $0.62 | No | $0.00 | 0.70% | 1 | 0.019 |
| Horton Park | $5.27 | Yes | $5.27 | 9.20% | 14 | 0.144 |
| Imperial River Boat Ramp | $0.42 | No | $0.00 | 3.30% | 5 | 0.012 |
| Koreshan State Historic Site | $0.36 | No | $0.00 | 0.70% | 1 | 0.011 |
| Punta Rassa Boat Ramp | $1.27 | Yes | $1.27 | 9.80% | 15 | 0.037 |
| Sanibel Island | $0.73 | Yes | $0.73 | 2.60% | 4 | 0.022 |
| Bonita Beach Resort Motel | $0.09 | No | $0.00 | 0.00% | 0 | 0.003 |
| Cape Harbour Marina | $0.77 | Yes | $0.77 | 1.30% | 2 | 0.023 |
| Ramp on Ohio Avenue | $0.24 | Yes | $0.24 | 0.00% | 0 | 0.007 |
| Castaways Marina | $1.07 | No | $0.00 | 0.00% | 0 | 0.029 |
| Tween Waters Marina | $1.04 | No | $0.00 | 2.60% | 4 | 0.026 |
| Mullock Creek Marina | $0.28 | No | $0.00 | 5.20% | 8 | 0.008 |
| Fish Trap Marina | $0.20 | No | $0.00 | 0.00% | 0 | 0.006 |
| Riverside Park | $0.11 | No | $0.00 | 0.00% | 0 | 0.003 |
| Pine Island Marina | $0.51 | No | $0.00 | 0.00% | 0 | 0.015 |
| Leeward Yacht Club #2 | $0.21 | Yes | $0.21 | 0.00% | 0 | 0.006 |
| Russell Park Ramp | $0.09 | Yes | $0.09 | 0.00% | 0 | 0.003 |
| Burnt Store Marina | $0.17 | No | $0.00 | 1.30% | 2 | 0.005 |
| Pineland Marina | $0.51 | No | $0.00 | 2.00% | 3 | 0.015 |
| Terra Verde Country Club | $0.16 | No | $0.00 | 0.00% | 0 | 0.005 |
| No Name Ramp | $0.21 | Yes | $0.21 | 0.00% | 0 | 0.007 |
| Jug Creek Cottages | $1.49 | No | $0.00 | 7.20% | 11 | 0.044 |
| Monroe Canal Marina | $0.78 | Yes | $0.78 | 5.90% | 9 | 0.023 |
| Viking Marina | $9.15 | No | $0.00 | 19.60% | 30 | 0.236 |
| Hickory Bait and Tackle | $0.36 | No | $0.00 | 0.00% | 0 | 0.011 |
| Inlet Motel | $0.74 | No | $0.00 | 0.00% | 0 | 0.022 |
| TOTAL | $36.61 | | $14.63 | | | |

One caveat for the models we present of Lee County relates to the on-the-water site choice model. Because many of the water site variables are spatially correlated (due to the fact many on-the-water sites are in the same 12 square mile polygon), the model is not well suited to evaluating the effect of changes in individual on-the-water site characteristics. However, the model does perform well in terms of predicting water site choice, and hence, the model does a good job of predicting the utility index (inclusive value) of the available on-the-water sites from any ramp. Thus, the combined models are well suited to the valuation of ramps, but less-well suited to the valuation of changes in specific water site characteristics.

The model we present is based on boaters that have launched from ramps in Lee County. Thus, the scope of the model or what might be referred to as the "market area" covered by the model is boaters utilizing public ramps in Lee County. Lee County is a large area with many possible public ramps available to boaters. It is natural to think that ramps within Lee County are a part of the relevant market area for the segment of boaters that have used a Lee County ramp. These ramps are also natural substitute sites for Lee County boaters. Our model includes these possibilities. However, it may be that the geographic market area includes some ramps and boaters using other ramps outside of Lee County. For example, when the characteristics of a Lee County ramp are improved, it may attract some boaters that were not previously using a Lee County ramp. These boating behaviors occurring outside of Lee County would not be captured by these Lee County RUMs. In this case, our model may underestimate the benefits of a Lee County ramp improvement because it cannot capture the benefits to potential new users of Lee County ramps. That said, when improvement occurs, we know that the

main beneficiaries are those already using Lee County ramps and these benefits are captured by our models. Conversely, our model will overstate losses due to ramp closures because boaters cannot adjust their trips by going outside of Lee County or by reducing their total trips. This effect will be small for closures affecting sites with low visitation or sets of sites with relatively low visitation [35].

*Simulated Losses Due to HAB-Related Ramp Closures*

Given the noxious nature of cyanobacterial HABs, losses resulting from these events can be simulated as closures of sites that are impacted by blooms. To create a realistic scenario of site closures during a cyanobacterial HAB, we identify the boat ramps located in the low salinity Caloosahatchee River and differentiate them from those located in high salinity areas beyond the mouth of the river (Figure 1). Given cyanobacterial biology, HABs caused by these organisms can be expected to dissipate as the blooms reach high salinity areas. Therefore, HABs can be expected to cause actual or perceived loss of access from boat ramps located in the river, but access will remain relatively untouched in ramps located in the Gulf of Mexico or in high salinity areas beyond the river mouth.

Therefore, losses accruing to recreational boaters as a result of cyanobacterial HABs in Lee County can be estimated as the difference between the CS with the full choice set and the CS with the restricted choice set where the boat ramps in the river are assumed to be closed (Equation (11)). To approximate this, we can sum up the losses associated with individual site closures. As shown in Table 3, lost value due to closure can be measured by the sum of the lost CS per choice occasion over the full choice set, which was estimated to be $14.63 (2009 dollars, or $17.26 2018 dollars). This approximation undervalues the loss because it does not account for the fact that multiple sites were closed at once nor does it include any losses in fixed costs during short-term closures [8], yet the approximation could overvalue the losses since the model does not allow for substitution out of boating or out of Lee County. Nevertheless, the approach does illustrate how to easily derive a reasonable estimate of losses if one does not have access to the underlying data needed to compute Equation (11). In other words, the average individual who would have taken a boating trip to Lee County on a day when cyanobacterial HABs were occurring in 2018 is estimated to have lost $17.26 as a direct result of the algae blooms.

An aggregate loss for the months when HABs occur can be estimated by multiplying the $17.26 per choice occasion loss in consumer surplus as a result of the cyanobacterial HABs by the number of Lee County trips that would normally have occurred in those months if there were no HABs. Table 4 provides aggregate loss estimates for each month of the year, assuming that the algae bloom lasts for the entire duration of the month. These monthly figures take into account seasonal recreation patterns and expected annual growth in trips (Figure 4, [9]).

**Table 4.** Consumer surplus lost in closure scenarios for 12 months (2018 dollars).

| Month of Closure | Estimated Loss |
| --- | --- |
| January | $729,752.64 |
| February | $624,098.22 |
| March | $653,583.17 |
| April | $928,776.09 |
| May | $948,432.72 |
| June | $938,604.41 |
| July | $992,660.16 |
| August | $1,041,801.75 |
| September | $1,592,187.58 |
| October | $1,253,110.59 |
| November | $914,033.61 |
| December | $835,407.06 |

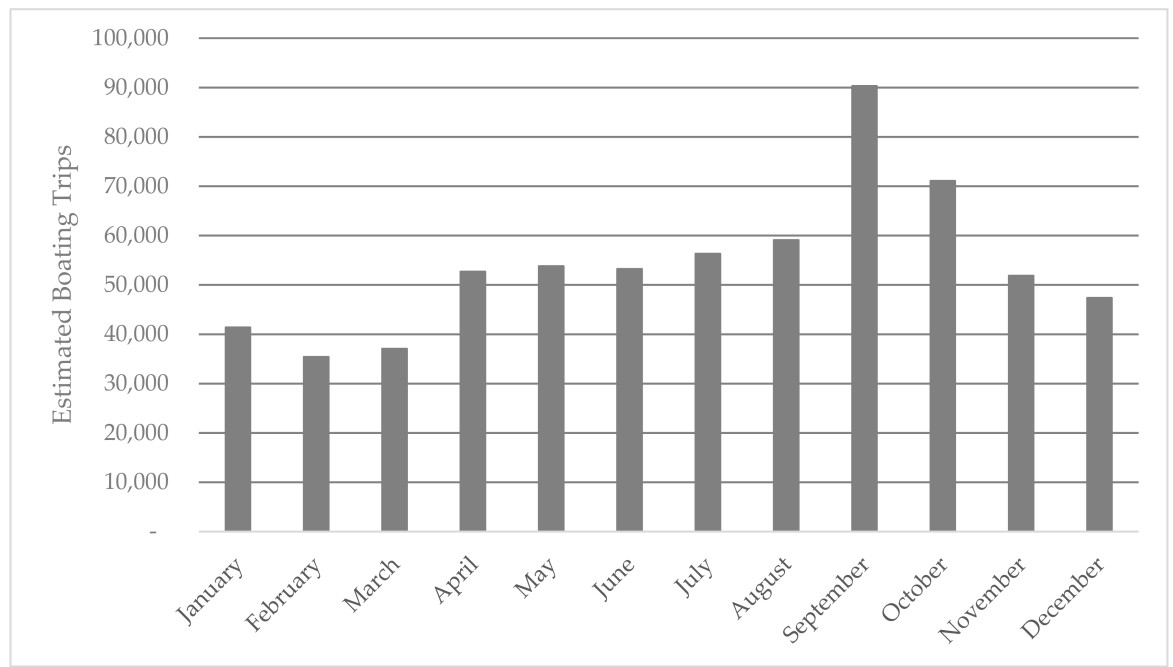

**Figure 4.** Estimated monthly number of boating trips to Lee County. Adapted from FWC [9].

This framework can be used to estimate losses from historic or hypothetical algae blooms in Lee County. For example, the 2018 blooms were first documented at the end of June, and relatively high concentrations of the cyanotoxin microcystin were documented through July, August, and September, with relatively lower concentrations documented all the way to the end of the year [27]. Thus, losses for the 2018 cyanobacterial HAB in Lee County can be estimated at $3,550,537 (2018 dollars), or the sum of the monthly estimates for July, August, and September. Losses for hypothetical (or observed) HAB scenarios of different durations can be estimated using Table 4 along with the hypothetical (or observed) duration of the HABs.

## 6. Discussion

While the model provides credible estimates of the losses to recreational boaters as a result of the cyanoHABs, it is limited in several accounts. Estimates produced by this model are biased in both directions. On one hand, the model underestimates since (1) the site-specific CS estimates in Table 3 do not jointly evaluate the closure, and, since (2) they do not account for any losses in fixed costs due to short-term closures [8]. On the other hand, the model overestimates since (1) discontinuing participation in Lee County boating is not a substitution option in the model, and since (2) there may be days where HABs cause a decline in site quality but do not fully close a ramp. The assumption that HABs result in full closure is similar to the assumption made by [39] to illustrate how to use benefit transfer to estimate damages of HABs on Lake Erie.

The per choice occasion welfare loss of $17.26 reported in this study is in the range of others previously estimated and reported in the literature. For instance, [32] has shown that when reviewing the effects of the BP oil spill in the Gulf of Mexico, losses have been reported as low as $2.23 for individuals fishing from private boats, but as high as $34.27 for individuals fishing from chartered boats. Similarly, [33] reports welfare estimated losses to shoreline recreationists of between $37.23 and $40.41 per lost trip as a result of the BP oil spill.

It is important to note that while these results provide an estimate of some losses in ecosystem services arising from HABs, they do not provide a complete picture of all losses in ecosystem services resulting from these events. Our estimates only include losses to recreational boaters, and do not account for losses in other cultural, provisioning, regulating or supporting ecosystem services that are

impacted by cyanobacterial HABs. Thus, losses from HABs in Lee County can be expected to be larger than the estimates provided here.

Future research efforts aimed at better understanding the impact of HABs on recreational services of coastal and marine ecosystems could focus on estimating the demand response to HABs that are near a ramp but do not close a ramp. These estimates can be obtained using stated preference surveys [12] or by developing estimates based on empirical observations of changes in behavior resulting from changes in the quality of ecosystem services [32,33], rather than on simulations using previously collected data (such as the estimates developed in this study).

The approach developed in this study could be coupled with biophysical models of the movement of HABs or other hazards that negatively impact the provision of cultural ecosystem services to develop quasi-real-time damage estimates as a result of these hazards. For instance, a biophysical model of the movement of HABs along the Caloosahatchee River could be used in tandem with the site- or ramp-based approach developed here to simulate closures in individual boat ramps as the cyanobacterial blooms flow from Lake Okeechobee to the Gulf of Mexico. Similar site-based closure simulations could be applied to other settings where movement of hazards is relatively predictable.

## 7. Conclusions

A RUM model was used to compute the value of changing site characteristics as well as to estimate the value of access for available recreational boating sites in Lee County. As expected, the more popular ramps have higher per choice occasion values, but all sites have fairly similar values per-trip to the site. The model was also applied to assess the loss of cultural ecosystem services in terms of foregone recreational boating opportunities as a result of harmful algae blooms. Losses to recreational boating resulting from the 2018 blooms in Lee County are estimated at $3.5 million (2018 dollars). This approach can be adapted to other contexts where ecological disturbances are causing closures of recreational areas.

A.    The Millennium Ecosystem Assessment (MEA 2005) defines cultural ecosystem services as nonmaterial benefits that are enjoyed through recreation and aesthetic experiences, spiritual or artistic appreciation.

B.    Travel costs were assumed to be $0.50 per mile in 2009 dollars (equivalent to $0.59 2018 dollars).

C.    Travel times for two sites (Sanibel and Lovers Key) were adjusted downward to 20 mph for a portion of their travel distance to account for slower speeds on causeways and highly congested areas.

**Supplementary Materials:** The following are available online at http://www.mdpi.com/2073-4441/11/6/1250/s1, Data: KML file with boat ramp locations that indicates whether they are open or closed in our HAB scenario.

**Author Contributions:** Conceptualization, S.A., F.L., D.S., M.T.; methodology, S.A., F.L., D.S., M.T.; software, F.L.; validation, F.L., M.T.; formal analysis, S.A., F.L.; investigation, F.L., M.T.; resources, S.A., F.L., D.S., M.T.; data curation, F.L, M.T..; writing—original draft preparation, S.A., D.S.; writing—review and editing, S.A., F.L., D.S., M.T.; visualization, S.A.; supervision, S.A., F.L., D.S., M.T.; project administration, M.T.; funding acquisition, M.T.

**Funding:** The initial phase of this research was funded by the Florida Fish and Wildlife Conservation Commission, grant number FWC 04/05-23.

**Acknowledgments:** The support of the Florida Fish and Wildlife Conservation and Lee County are gratefully acknowledged. We thank Dave B. Harding and Ed Mahoney for their input in the initial phase of research.

**Conflicts of Interest:** The authors declare no conflict of interest. The funders had no role in the design of the study; in the collection, analyses, or interpretation of data; in the writing of the manuscript, or in the decision to publish the results.

## Appendix A

**Table A1.** Latitude and longitude coordinates of boat ramp groups in the study.

| Ramp Name | Lat. | Long. |
|---|---|---|
| BMX Strausser | 26.62438 | −81.98989 |
| Alva Boat Ramp | 26.71398 | −81.60597 |
| Burnt Store Boat Ramp | 26.64714 | −82.04193 |
| Cape Coral Yacht Basin | 26.54266 | −81.95226 |
| Lovers Key/Carl E. J | 26.39354 | −81.86658 |
| City of Fort Myers Yacht Basin | 26.64579 | −81.87249 |
| Ft Myers Shores Davis Ramp | 26.71155 | −81.75297 |
| Franklin Locks North | 26.72375 | −81.69232 |
| Franklin Locks South | 26.72098 | −81.69445 |
| Bokeelia Boat Ramp | 26.69431 | −82.1459 |
| Horton Park | 26.60736 | −81.91359 |
| Imperial River Boat Ramp | 26.33863 | −81.80495 |
| Koreshan State Historic Site | 26.43663 | −81.81962 |
| Punta Rassa Boat Ramp | 26.48458 | −82.01049 |
| Sanibel Island | 26.45354 | −82.03592 |
| Bonita Beach Resort Motel | 26.35391 | −81.85499 |
| Cape Harbour Marina | 26.54383 | −82.00788 |
| Ramp on Ohio Avenue | 26.45115 | −81.94748 |
| Castaways Marina | 26.48152 | −82.18026 |
| Tween Waters Marina | 26.5109 | −82.18947 |
| Mullock Creek Marina | 26.47347 | −81.8516 |
| Fish Trap Marina | 26.3306 | −81.83171 |
| Riverside Park | 26.34232 | −81.77971 |
| Pine Island Marina | 26.59273 | −82.12625 |
| Leeward Yacht Club #2 | 26.68775 | −81.79311 |
| Russell Park Ramp | 26.68295 | −81.81462 |
| Burnt Store Marina | 26.76042 | −82.05518 |
| Pineland Marina | 26.66218 | −82.15527 |
| Terra Verde Country Club | 26.49037 | −81.85468 |
| No Name Ramp | 26.67372 | −81.90007 |
| Jug Creek Cottages | 26.70388 | −82.15735 |
| Monroe Canal Marina | 26.50525 | −82.08285 |
| Viking Marina | 26.6354 | −82.06264 |
| Hickory Bait and Tackle | 26.39939 | −81.84127 |
| Inlet Motel | 26.7311 | −82.21302 |

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
