# Peer review of "Valuing Provision Scenarios of Coastal Ecosystem Services: The Case of Boat Ramp Closures Due to Harmful Algae Blooms in Florida"

_water, doi:10.3390/w11061250_

Round 1

Reviewer 1 Report

The authors set forth a case study of the potential impacts of cyanobacterial blooms (harmful algal blooms) on recreational boating along the Florida coast. The paper describes the impacts in terms of loss of one ecosystem service. The paper provides a sound rationale and well-referenced introduction, data, and well-reasoned conclusions. The figures and tables are appropriate. There are a few concerns. On line 113, "phosphorus" is mis-spelled, and the logic on lines 334-337 seems wrong: the word "cannot" on line 335 probably should be "can"? However, over-stating the losses is most likely correct. Also, when writing Lee County, the "c" should be upper case as it is a specific county and part of a proper name.

One issue that this reviewer noticed was the likely over-statement of the impact of salt on cyanobacteria. Cyanobacteria generally have a high tolerance for salt, part of their wide range of habitats in which they can thrive or survive. This may affect the numbers of ramps considered in the study. Nevertheless, it does not necessary impact the findings or the logic of the paper.

Generally, the paper is recommended for publication as submitted.

Author Response

The authors set forth a case study of the potential impacts of cyanobacterial blooms (harmful algal blooms) on recreational boating along the Florida coast. The paper describes the impacts in terms of loss of one ecosystem service. The paper provides a sound rationale and well-referenced introduction, data, and well-reasoned conclusions. The figures and tables are appropriate. There are a few concerns.

On line 113 , "phosphorus" is mis-spelled,

Spelling has been corrected.

and the logic on lines 334-337 seems wrong: the word "cannot " on line 335 probably should be "can"? However, over-stating the losses is most likely correct.

The sentence in paper is correct.  The model will overstate losses from a closure because it forces a boater to keep using a ramp in Lee County, even though their true next best alternative may be to stop boating or to go boating outside Lee County.  In other words, the model cannot capture the benefits of improving boat ramps in Lee County to individuals who were not using Lee County boat ramps at the time of the study. To improve clarity, we modified the wording of the sentence to replace “no participation in boating…” with “discontinuing participation in Lee County boating…”

Also, when writing Lee County  , the "c" should be upper case as it is a specific county and part of a proper name.

We have corrected these throughout.  Thank you for pointing this out.

One issue that this reviewer noticed was the likely over-statement of the impact of salt on cyanobacteria. Cyanobacteria generally have a high tolerance for salt, part of their wide range of habitats in which they can thrive or survive. This may affect the numbers of ramps considered in the study. Nevertheless, it does not necessary impact the findings or the logic of the paper.

While this may be true of cyanobacteria in general, the cyanobacteria blooms that have become prevalent in Florida are mostly of the genus Microcystis and Anabaena.  These types of cyanobacteria are very sensitive to salinity, and several studies (some of which are cited in the paper) have shown that their cell membranes cannot tolerate higher salinities.  We have added a figure (now labeled Figure 2) showing water sampling data from 2018, when the latest HABs took place in Lee County.  The figure shows that concentrations of microcystin (the toxin produced by the cyanobacteria prevalent in Florida) were high in the low salinity areas in the river, but very low or not present in areas outside the river mouth.  As mentioned in the paper, we used these data to create our HAB scenarios.

Generally, the paper is recommended for publication as submitted.

Thank you for very helpful comments.

Reviewer 2 Report

The article is quite interesting and the topic is worthy of investigation. The methodology is rigorous and results are relevant and seem to be supported by the analysis. Nevertheless, several important shortcomings have been found in different sections of the paper such the literature review, the discussion or the conclusions. In addition, other minor problems are also found all over the text. Therefore I should recommend a major revision before considering the article for publication. Below I detail the main and minor issues detected in the manuscript:

INTRODUCTION AND LITERATURE REVIEW 

The review of the state of the art in the manuscript is very scarce, in fact it hardly appears in the text. In essence, the authors go directly through the exposure of their specific case study since the beginning of the text (other issues in the introduction such as the justification of the relevance of the topic or the need to fill the with this research gaps existing in this scientific field are also quite scarce). The citation of other cases studies, methodologies, etc. to justify the relevance of the topic and review the state of the art in a comprehensive way within this scientific field is missing in text. The reference to recent research studies in international scientific journals of relevance is also missing (the bibliography is mainly focused to local references and books, and articles with a more than 15 years old, in a field of research that has undergone great innovations in the last 10 years). I recommend the authors to include at least 7-8 recent relevant publications from international scientific journals from other case studies, different methodologies, etc. to illustrate the topic and its relevance (e.g. two interesting case studies to cite may these ones:

- Impacts on cultural ecosystem services because of algae bloom in UK in Willis, Ch. et al. Harmful algal blooms: the impacts on cultural ecosystem services and human well-being in a case study setting, Cornwall, UK, Marine Policy 97, 2018, 232-238. 

- GIS tourism ecosystem services loss in a Spanish lagoon due to eutrophication process in Garcia-Ayllon, S. The Integrated Territorial Investment (ITI) of the Mar Menor as a model for the future in the comprehensive management of enclosed coastal seas. Ocean Coast. Manag. 2018, 166, 82–97.)

METHODOLOGY AND RESULTS 

Authors should state the software used for GIS and numerical analysis. In addition, try to make the text more understandable for readers that do not belong to Florida (take into account that Water is an international journal with readers all over the world, so avoid very specific data not explained in detail in the text or names of places or areas that are not cited in a explanatory map in the manuscript for example).

DISCUSSION 

The discussion section is quite short and disappointing.  Actually it is not a scientific discussion, but rather a brief summary of the article, which is not relevant in this part of the text (I recommend the authors to carefully read the template provided in the journal to find out what should be the content of this section). The authors must at least contrast here the relevance and validity of the results obtained by comparing them with other studies and research, observing possible deficiencies of their methodology, and proposing possible improvements to be included in future research lines for example. In this case, since there are no other studies in the same area to compare, it may be interesting to compare results of similar studies using other economic monetization methodologies for the analysis of the loss of ecosystem services related to the environmental impact of tourism in other places of the  world. For example, which advantages/similarities/differences offer the RUM approach of authors from well known and relevant such the WTP approach? Do the conclusions converge from a conceptual point of view with those obtained in other studies? (compare for example with other similar studies with WTP Turnbull and logit/probit economical analysis of environmental issues in Schumann et al. in Barbados beach in Visitors’ willingness to pay marine conservation fees in Barbados. Tour. Manag. 2019, 71, 315-326 or in Mar Menor Mediterranean lagoon in García-Ayllón, S. New Strategies to Improve Co-Management in Enclosed Coastal Seas and Wetlands Subjected to Complex Environments: Socio-Economic Analysis Applied to an International Recovery Success Case Study after an Environmental Crisis. Sustainability 2019, 11, 1039). 

CONCLUSIONS

There are no conclusions in the manuscript. Author should at least include a conclusive text in the manuscript in order that readers of the journal can easily assess their interest in the article at a glance by only reading the abstract and the conclusions. Given the complexity of methodology and results I suggest the authors not to merge this conclusive text in the discussion one, but rather to include a specific section.

MINOR ISSUES 

- Figure 1 and Figure 1A are not very illustrative (in my opinion it may be more interesting to number the places in Fig 1 and, instead of providing their coordinates in Fig1A, to include a KML file as supplementary data to make the readers an easy access to information).

- Table such as Table 3 must be better formatted to fit in the paper

- The paragraphs must be formatted correctly on the right side of the paper.

Author Response

Comments and Suggestions for Authors

The article is quite interesting and the topic is worthy of investigation. The methodology is rigorous and results are relevant and seem to be supported by the analysis. Nevertheless, several important shortcomings have been found in different sections of the paper such the literature review, the discussion or the conclusions. In addition, other minor problems are also found all over the text. Therefore, I should recommend a major revision before considering the article for publication. Below I detail the main and minor issues detected in the manuscript:

Thank you for this very thoughtful and helpful review.

INTRODUCTION AND LITERATURE REVIEW

The review of the state of the art in the manuscript is very scarce, in fact it hardly appears in the text. In essence, the authors go directly through the exposure of their specific case study since the beginning of the text (other issues in the introduction such as the justification of the relevance of the topic or the need to fill the with this research gaps existing in this scientific field are also quite scarce). The citation of other cases studies, methodologies, etc. to justify the relevance of the topic and review the state of the art in a comprehensive way within this scientific field is missing in text. The reference to recent research studies in international scientific journals of relevance is also missing (the bibliography is mainly focused to local references and books, and articles with a more than 15 years old , in a field of research that has undergone great innovations in the last 10 years). I recommend the authors to include at least 7-8 recent relevant publications from international scientific journals from other case studies, different methodologies, etc. to illustrate the topic and its relevance (e.g. two interesting case studies to cite may these ones:

- Impacts on cultural ecosystem services because of algae bloom in UK in Willis, Ch. et al. Harmful algal blooms: the impacts on cultural ecosystem services and human well-being in a case study setting, Cornwall, UK, Marine Policy 97, 2018, 232-238.

- GIS tourism ecosystem services loss in a Spanish lagoon due to eutrophication process in Garcia-Ayllon, S. The Integrated Territorial Investment (ITI) of the Mar Menor as a model for the future in the comprehensive management of enclosed coastal seas. Ocean Coast. Manag. 2018, 166, 82–97.)

In addition to the suggested references, we have added the following:

Dyson, K., Huppert, D.D. Regional economic impacts of razor clam beach closures due to harmful algal blooms (HABs) on the Pacific coast of Washington. Harmful Algae, 2010, 9, 3, 264-271.

Backer, L.C. Impacts of Florida red tides on coastal communities. Harmful Algae, 2009, 8, 4, 618-622.

Hoagland, P., Jin, D., Beet, A., Kirkpatrick, B., Reich, A., Ullmann, S., Fleming, L.E., Kirkpatrick, G. The human health effects of Florida Red Tide (FRT) blooms: An expanded analysis. Environment International, 2014, 68, 144-153.

Morgan, K.L., Larkin, S.L., Adams, C.M. Red tides and participation in marine-based activities: Estimating the response of Southwest Florida residents. Harmful Algae, 2010, 9, 3, 333-341.

In addition, we have added three more references in the methodology section:

Freeman, A.M., J.A. Herriges, and C.L. Kling. The Measurement of Environmental and Resource Values: Theory and Methods, third edition, RFF Press, 2014.

Parsons, G.R. Travel cost models. In: A Primer on Nonmarket Valuation, second edition, Champ, P.A., K.J. Boyle, and T.C. Brown, editors, Springer, 2017.

English, E. R.H. von Haefen, J. Herriges, C. Leggett, F. Lupi, K. McConnell, M. Welsh, A. Domanski, and N. Meade. Estimating the Value of Lost Recreation Days from the Deepwater Horizon Oil Spill. Journal of Environmental Economics and Management, 2018, 91, 26–45.

METHODOLOGY AND RESULTS

Authors should state the software used for GIS and numerical analysis . In addition, try to make the text more understandable for readers that do not belong to Florida  (take into account that Water is an international journal with readers all over the world, so avoid very specific data not explained in detail in the text or names of places or areas that are not cited in a explanatory map in the manuscript for example).

Thank you for this comment.  We have prepared a map to help readers who are not familiar with Florida that shows all the features identified in the text, as well as other features such as cities that were not previously mentioned (now included as Figure 1).  We have also noted in the manuscript that the software used for GIS is ArcMap in the ArcGIS suite, and the software used for numerical analysis is LIMDEP.

DISCUSSION

The discussion section is quite short and disappointing.  Actually it is not a scientific discussion, but rather a brief summary of the article, which is not relevant in this part of the text (I recommend the authors to carefully read the template provided in the journal to find out what should be the content of this section). The authors must at least contrast here the relevance and validity of the results obtained by comparing them with other studies and research, observing possible deficiencies of their methodology, and proposing possible improvements to be included in future research lines for example. In this case, since there are no other studies in the same area to compare, it may be interesting to compare results of similar studies using other economic monetization methodologies for the analysis of the loss of ecosystem services related to the environmental impact of tourism in other places of the  world. For example, which advantages/similarities/differences offer the RUM approach of authors from well known and relevant such the WTP approach? Do the conclusions converge from a conceptual point of view with those obtained in other studies? (compare for example with other similar studies with WTP Turnbull and logit/probit economical analysis of environmental issues in Schumann et al. in Barbados beach in Visitors’ willingness to pay marine conservation fees in Barbados. Tour. Manag. 2019, 71, 315-326 or in Mar Menor Mediterranean lagoon in García-Ayllón, S. New Strategies to Improve Co-Management in Enclosed Coastal Seas and Wetlands Subjected to Complex Environments: Socio-Economic Analysis Applied to an International Recovery Success Case Study after an Environmental Crisis. Sustainability 2019, 11, 1039).

The discussion section has been thoroughly re-worked to address the shortcomings identified by the reviewer.  The section begins by highlighting caveats or deficiencies of the approach. A comparison of welfare estimates from this study with studies estimating similar welfare losses for the BP oil spill in the Gulf of Mexico follows. We close the section with a discussion of avenues for future research.  However, the suggested studies are not really comparable, as many of them don’t really estimate economic losses—for instance Garcia-Ayllon does not provide any quantitative economic estimates so we cannot make a comparison; similarly, studies simply estimating the value of recreation that do not measure a loss in this value as a result of environmental damage or a reduction in environmental quality are less relevant points of comparison.

CONCLUSIONS

There are no conclusions in the manuscript. Author should at least include a conclusive text in the manuscript in order that readers of the journal can easily assess their interest in the article at a glance by only reading the abstract and the conclusions. Given the complexity of methodology and results I suggest the authors not to merge this conclusive text in the discussion one, but rather to include a specific section.

We have created a separate but short conclusion section as suggested.

MINOR ISSUES

- Figure 1 and Figure 1A are not very illustrative (in my opinion it may be more interesting to number the places in Fig 1    and, instead of providing their coordinates in Fig1A, to include a KML file as supplementary data to make the readers an easy access to information).

We appreciate and fully considered the reviewer’s feedback, but we have opted to retain the original figure (now Figure 3). We have, however, created the suggested supplementary KML data file that labels each ramp as well as indicated whether they are open or closed in our HAB scenario.

- Table such as Table 3 must be better formatted to fit in the paper

We have made some additional minor format changes and have spent significant effort trying to make this table look good, but the template is quite limiting with what we can do with a table of this size.

- The paragraphs must be formatted correctly on the right side of the paper.

Paper has been formatted using the justify option as requested.

Round 2

Reviewer 2 Report

The authors have globally implemented/answered to all the suggestions made in my previous report and the manuscript has been improved in this new version. In my opinion, it can be considered for publication now.